# Neuroprotective Effect of Melatonin on Sleep Disorders Associated with Parkinson’s Disease

**DOI:** 10.3390/antiox12020396

**Published:** 2023-02-06

**Authors:** Xinyu Hu, Jingwen Li, Xinyi Wang, Hanshu Liu, Tao Wang, Zhicheng Lin, Nian Xiong

**Affiliations:** 1Department of Neurology, Union Hospital, Tongji Medical College, Huazhong University of Science and Technology, Wuhan 430022, China; 2Laboratory of Psychiatric Neurogenomics, McLean Hospital, Harvard Medical School, Belmont, MA 02478, USA

**Keywords:** Parkinson’s disease, melatonin, sleep disorders, mechanisms, clinical trials

## Abstract

Parkinson’s disease (PD) is a complex, multisystem disorder with both neurologic and systemic manifestations, which is usually associated with non-motor symptoms, including sleep disorders. Such associated sleep disorders are commonly observed as REM sleep behavior disorder, insomnia, sleep-related breathing disorders, excessive daytime sleepiness, restless legs syndrome and periodic limb movements. Melatonin has a wide range of regulatory effects, such as synchronizing circadian rhythm, and is expected to be a potential new circadian treatment of sleep disorders in PD patients. In fact, ongoing clinical trials with melatonin in PD highlight melatonin’s therapeutic effects in this disease. Mechanistically, melatonin plays its antioxidant, anti-inflammatory, anti-excitotoxity, anti-synaptic dysfunction and anti-apoptotic activities. In addition, melatonin attenuates the effects of genetic variation in the clock genes of Baml1 and Per1 to restore the circadian rhythm. Together, melatonin exerts various therapeutic effects in PD but their specific mechanisms require further investigations.

## 1. Introduction

Dysregulation of the circadian rhythm and sleep-wake may occur in various diseases, such as Parkinson’s disease (PD) [1]. As increasingly recognized, the functional incoordination of multiple parts of both central and peripheral systems and diversified neurotransmitters is affected for sleep disorders as non-motor symptoms in patients with PD [2]. PD is an extra-pyramidal disease with a slightly higher incidence in men than in women, affecting more than 1 in every 100 people aged over 60 [3]. The pathological hallmark of PD is dopaminergic neuronal death, which may affect 60% of total dopaminergic neurons [4]. A fundamental abnormality in PD is the accumulation of α-synuclein in the forms of Lewy bodies, which mediates the cell death [5]. The progressive loss of dopamine neurons makes the PD patients suffer from a large number of motor and non-motor features which can affect their health to a variable degree [4]. Motor symptoms of PD are characterized by quiescent tremor, slow movement, increased muscle tension and postural instability [4]. Additionally, non-motor symptoms are an integral component of PD, including sleep disorders, mood disturbances, cognitive impairments and apathy as the common ones [6].

Sleep disorders are among the most common non-motor manifestations in PD and have a significantly negative impact on quality of life [7]. There are different types of sleep disorders in PD, including rapid eye movement (REM), sleep behavior disorder (RBD), insomnia, sleep related breathing disorders (SBDs), excessive daytime sleepiness (EDS), restless legs syndrome (RLS) and periodic limb movements (PLM) among others [8].

Management of these sleep disorders often require complex therapeutic regimens, involving both pharmacological and non-pharmacological interventions [9]. For both health and wellbeing, comprehensive treatment of PD is quite essential; therefore that a great deal of clinical trials have been carried out to verify the efficacy of these interventions [10]. Nonetheless, these therapy methods often have a series of side effects, including excessive daytime sleeping, cognitive impairment, poor tolerance and so on; moreover, long-term use may also lead to drug dependence [11]. As one of the pharmacological interventions, melatonin (namely N-acetyl-5-methoxytryptamine), is well known for its natural synthesis within the pineal gland and reportedly used to treat sleep disorders, especially for RBD and insomnia in PD [12]. 

However, both efficiency and mechanisms by which melatonin treats the sleep disorders in PD need to be investigated and elucidated. Here we review the field in four topics, (1) PD and melatonin, (2) sleep disorders of PD and their managements and (3) possible neuroprotective mechanisms of melatonin therapy for sleep disorders in PD. We aim to provide the current state of melatonin-based interventions, including what we have learned and what are the needs in the intervention to sleep disorders in PD patients.

## 2. PD and Melatonin

Melatonin is a neurohormone with chronobiological effects that control circadian rhythms [13]. It was first described in 1958 by a dermatologist named Aaron Lerner [14], secreted mainly from the pineal gland situated at the center of the brain. It has a wide range of regulatory and protective effects, such as synchronizing circadian rhythm, protecting against oxidative stress, regulating energy metabolism, modulating the immune system, and postponing the ageing process [15]. The content of melatonin is quite little, as estimated between 10–80 mg per night in the body, the bottom-most values for a hormone secretion [13]. Its biosynthesis gradually declines with age, reduced by 10%~15% for every 10 years on average, especially after 35 [13]. Melatonin is released into the bloodstream exclusively at night following the circadian rhythm [16]. Alteration in circadian melatonin production has been reported in neurodegenerative diseases [17]. The secretion of melatonin is a key signal for sleep–wake cycle organization and has relevant neuroprotective activity in a number of experimental models [18]. Melatonin facilitates achieving better sleep for these patients by reducing the sleep-onset latency or by regulating sleep–wake times to coincide with the natural circulatory cycle, as well as reducing sleep episodes without muscle atonia [19].

In recent years, studies on its biological functions have shown that melatonin has many physiological functions, such as promoting sleep, regulating jet lag, anti-aging, regulating immunity and anti-tumor [20]. Furthermore, melatonin has preventive and therapeutic effects for many neurological disorders, including PD, Alzheimer’s disease, multiple sclerosis, etc. [21]. Melatonin exerts its function by binding to two main receptors, MT1 and MT2 [22]. Adi, N et al. determined the MT1 and MT2 receptors’ expressions in whole brain post-mortem tissue from the amygdala and substantia nigra (SN) of well-characterized PD and non-neurologic control subjects by the real-time polymerase chain reaction (PCR) [23]. They found that PD cases showed a statistically significant decrease in the MT1 receptor expression in both SN and the amygdala versus normal controls. The expression of the MT2 receptor was also decreased in both SN and the amygdala versus normal controls. The results demonstrated a down-regulation of melatonin receptors in regions affected by PD, indicating the relationship between melatonin and PD to some extent. Tamtaji, O.R et al. used the rotenone-induced PD male Wistar rat model to understand circadian dysfunction in PD, and then separated them into two groups: one is rotenone and melatonin; the other is a rotenone and melatonin vehicle [12]. The results showed that melatonin could stop the rotenone-induced phase alteration in rat Cry1 (rCry1) daily rhythm. Preclinical and clinical studies have shown that melatonin supplementation is an appropriate therapy for PD, especially for the sleep disorders [19].

Despite the multifactorial etiology, the pronounced decline in nocturnal melatonin synthesis is common in PD patients [21]. They exhibit not only reduced amounts of secreted melatonin, but also a higher degree of irregularities in melatonin production. Therefore, the melatonin rhythm has lost not only signal strength in clock resetting, but also its reliability as an internal synchronizing time cue [21]. Loss or damage of the neurons in the SN and other parts of the circadian timing system may account for the circadian rhythm abnormalities and sleep disorder seen in PD patients [24]. To inspect the potential of melatonin therapy in older patients with sleep disorders, Haimov, I and P. Lavie set a run-in period (where no treatment was administered) and four experimental periods [25]. They found that 1 week’s treatment of 2 mg of fast-release melatonin was as effective as 2 months’ treatment with 1 mg of sustained-release melatonin, on sleep initiation. The results convincingly demonstrated that melatonin could increase sleep efficiency in elderly insomniacs by decreasing nighttime activity.

In addition to melatonin, melatonin analogues are also promising therapeutical approaches for PD [26]. Recently developed pharmacological agents, such as ramelteon, tasimelteon and agomelatine are melatonin receptor agonists which, compared with melatonin itself, have a longer half-life time and greater affinity for the melatonin receptors [26]. Consequently, they are thought to hold promise for treating a variety of sleep disorders. Ramelteon is a novel melatonin receptor agonist that has been shown to act on both MT1 and MT2 receptors, and has a longer duration of action than melatonin itself [27]. As the first commercially available melatonin receptor agonist in 2005, ramelteon could mimic the physiological effects of melatonin [28]. The participation of receptors MT1 and MT2 can affect the maintenance of the normal sleep–wake cycle [29]. It can be used for the treatment of patients with difficulty to fall asleep or with chronic and transient insomniacs, through reducing the sleep latency and improving the daytime sleep [29]. Tasimelteon, another MT1/MT2 agonist currently under evaluation, has also been shown effective for sleep re-synchronization [30].

Some authors, however, held a view that melatonin might play a deleterious role in impaired neurons with the process of dopamine degeneration, so that its antagonism might enhance recovery from PD conditions [31]. Dr. Willis proposed that ML-23 (N-[3,5-dinitrophenyl]-5-methoxytryptamine) and S-20928 (N-[2-(1-naphthyl) ethyl] cyclobutyl carboxamide) were used to antagonize the melatonin receptors and studied their pharmacological actions [31]. The mechanism involved in the repair effects observed with ML-23 could be mediated by ML-23’s ability to antagonize melatonin and to counteract the effect of melatonin on the cytoskeleton and impaired axoplasmic transport in dying neurons. Like the putative melatonin receptor antagonist ML-23, the antagonist S-20928 also seemed to possess anti-PD properties that enhanced the recovery in a chronic model of PD [31]. Moreover, it was thought that the combination of indole- and hydrazone-type compounds might provide new effective drugs against free radicals and give a new perspective to melatonin analogues [15]. 

## 3. Sleep Disorders Associated with PD and Their Management

Sleep disorders have been increasingly recognized with PD and can arise as a prodromal symptom, before the diagnosis of PD, even 15–20 years ahead of the well-known motor symptoms [2]. The patients may also suffer sleep disorders after the PD diagnosis, worsening the life quality of both patients and caregivers [8]. Randomized clinical trials examining the management of sleep disorders specifically in the PD population are rare while current treatments do not achieve satisfactory results [9]. Below, we will summarize the randomized controlled trials (RCTs) about non-melatonin-based managements and melatonin-based managements for five sleep disorders. These five sleep disorders are RBD, insomnia, SBDs, EDS and RLS/PLMS, which ordinarily precede or appear in advancing stages of PD (Table 1).

### 3.1. REM Sleep Behavior Disorder

RBD is a parasomnia, characterized by the loss of muscle atonia and abnormal behaviors during REM sleep (also known as RSWA) [32]. The prevalence of RBD has been estimated to be in the range of 0.5% to 2%, and between 5% and 13% of older community-dwelling adults aged 60 to 99 years [33]. Previously, the prevalence of RBD was estimated to range from 0.38% to 0.5% in general individuals [34], but it went up to 39–50% in PD patients based on analysis of data from multiple care centers defined by polysomnography [35]. Behaviors exhibited in RBD may place both the patient and bed partner at risk of physical harm, with between more severe cases involving strangulation and subdural hematoma [36]. It often occurs in neurological diseases, especially α-synuclein associated diseases, such as PD, dementia with Lewy bodies and multiple system atrophy [32,37]. 

Current treatment options for RBD involve supportive managements including altering the sleeping environment to make it safer (e.g., moving furniture and placing cushions next to the bed) [38]. Some randomized studies have been conducted on the treatment of RBD in patients with PD, containing benzodiazepines, cholinesterase inhibitor, MAO inhibitor drugs and 5-hydroxytryptophan (Table 1). In relation to the results, melatonin and clonazepam are the mainstay of pharmacological management [36,39]. Clonazepam, a long-acting benzodiazepine, has been widely considered as a first-line and highly effective therapy for RBD [40]. The recommended dose for clonazepam in the treatment of RBD is 0.25–3.0 mg taken 30–60 min before bedtime [41]. Yet, it has the potential disadvantages of exacerbating comorbid obstructive sleep apnea and cognitive impairment. Adverse effects of clonazepam may include sedation, sexual dysfunction and imbalance.

Some clinical evidence indicated that melatonin may be beneficial for RBD in PD patients. Building on several small case series of RBD patients treated successfully with up to 12 mg of melatonin, Dr. Kunz performed a double-blind, placebo-controlled, cross-over trial of melatonin. As a result, the RBD symptoms were improved in most cases [42]. It is not valid evidence for melatonin since the sample size was small, and only one was suffering from PD while eight consecutive outpatients were included. Subsequently, several clinical trials with greater sample sizes of melatonin and PD-RBD were performed. In some RCTs [43,44,45,46], doses of 3–4 mg of melatonin at bedtime were used successfully to reduce injuries (e.g., bruising, lacerations, fractures, and subdural hematomas) of PD patients caused by RBD, with few adverse effects. Melatonin’s mechanism of action against RBD remains unclear, but could be mediated by a combination of influences, including a direct impact on REM sleep atonia, stabilizing circadian clock variability and desynchronization, and increasing sleep efficiency [36]. Melatonin is generally well tolerated with minimal adverse events. These findings suggest that melatonin can be used in monotherapy or as an add-on therapy in selected patients with RBD and PD.

**Table 1 antioxidants-12-00396-t001:** RBD in Parkinson’s disease and their management supported by randomized controlled trials.

Method	Subjects (Dose/Duration)	Findings/Status	Study (Reference)
**Non-melatonin based management**			
Clonazepam	40 PD patients (0.5 mg, 4 weeks)	Failed to show superiority over the placebo.	Clinical trials (Shin C et al., 2019 [47])
	112 PD patients (1 mg, 4 weeks)	Significant improvement in RBDSQ scores and side effects were reported in three patients: two drowsiness and one dizziness.	Clinical trials (Hadi F, 2022 [43])
Cannabidiol	33 PD patients (75–300 mg, 12 weeks)	Showed no difference to the placebo for the mean total number of nights with events suggestive of RBD per week, according to sleep diary.	Clinical trials (de Almeida CMO, 2021 [48])
Safinamide	30 PD patients (50 mg, 3 months)	In 24/30 patients, the frequency and severity of abnormal RBD related motor behaviors improved, meanwhile, in 6/30 patients no substantial improvement was recorded concerning the relevant clinical symptoms.	Clinical trials (Plastino M, 2021 [49])
5-HTP	18 PD patients (50 mg, 4 weeks)	Significant improvement in the total percentage of stage REM sleep, but did not show significant reduction in frequency of RBD related episodes.	Clinical trials (Meloni M, 2022 [50])
Rivastigmine	12 PD patients (4.6 mg, 3 weeks)	Significantly reduced the mean frequency of RBD episodes, with occurring orthostatic hypotension and asthenia during the treatment.	Clinical trials (Di Giacopo R, 2012 [51])
**Melatonin based management**			
Melatonin	112 PD patients (3 mg, 4 weeks)	Significant improvement in RBDSQ scores and no adverse effects were reported in the melatonin group.	Clinical trials (Hadi F et al., 2022 [43])
	30 PD patients (4 mg, 8 weeks)	4 mg of melatonin is well-tolerated, but the number of RBD events after treatment was not reduced between melatonin groups and the placebo.	Clinical trials (Gilat M et al., 2020 [44])
	30 PD patients (2 mg, 4 weeks)	No statistically significant difference between melatonin and placebo groups regarding the RBDSQ decreasing.	Clinical trials (Ahn et al., 2020 [45])
	38 PD patients (3 mg, 6 weeks)	The melatonin-treated group had significant changes in total point scores on the PSG at the end.	Clinical trials (Litvinenko et al., 2012 [46])
	8 PD patients (2 of them with PD) (3–12 mg, 4 weeks)	Melatonin significantly reduced the number of 30-s REM sleep epochs without muscle atonia.	Clinical trials (Kunz D et al., 2010 [42])
Ramelteon ^1^	3 PD patients (8 mg, 8 weeks)	Terminated (Has results ^2^)	NCT00745030
**Differences**			
Melatonin is safer without drug dependence and with milder side effects.			

^1^ Melatonin’s analogues. ^2^ The trial had results but terminated eventually because of low subject recruitment and enrollment. However, the authors did not publish its detailed data. PD, Parkinson’s disease; REM, rapid eye movement; RBD, REM behavior disorder; RCT, randomized controlled trial; RBDSQ, RBD screening questionnaire; PSG, polysomnography; 5-HTP, 5-hydroxytryptophan.

### 3.2. Insomnia

Insomnia is defined by a repeated difficulty in sleep initiation, duration, consolidation or quality that occurs despite adequate time or opportunity for sleep [52]. Sleep fragmentation and sleep maintenance insomnia are among the most frequent sleep complaints reported by PD patients [53]. Insomnia symptoms are among the most common non-motor symptoms in PD and are their key determinants of quality of life [54]. Importantly, insomnia symptoms have been reported in up to 80% of individuals with PD [55]. Primary sleep disorders, PD-related motor symptoms and partial dopaminergic medications may bring about insomnia [52].

At present, the methods commonly used in the clinical treatment of insomnia in PD are divided into non-pharmacological strategies and pharmacological strategies (Table 2). Prior to embarking on pharmacologic or behavioral treatments for insomnia in PD patients, a careful evaluation of the type of ongoing sleep disturbance is needed [56]. In patients with chronic insomnia, regardless of etiology, treatment with behavioral interventions is recommended with the highest level of recommendation by the American Academy of Sleep Medicine [9]. Behavioral measures, collectively known as cognitive behavioral therapy for insomnia (CBT-I), may provide a strong evidence base in improving sleep quality in healthy individuals. However, the application of such measures is clearly contingent upon an individual patient and/or carer’s ability to implement/engage with them consistently, which may be impractical for many patients. Adherence is a key to efficacy [57]. 

In addition to the non-pharmacological strategies, a number of hypnotic agents, including benzodiazepines, Z-drugs and sedating antidepressants and antihistamines, are frequently used to treat insomnia [57]. Clinically, dopamine agonist is usually used [58]. Rotigotine has been reported to improve sleep quality and continuity in PD patients by promoting sleep stability and increasing REM [59]. However, potential risks of dopamine agonist-associated sleepiness, including sleep attacks, should be considered when prescribing dopaminergic therapies to improve overnight sleep [59]. Despite its lack of evidence for its efficacy, doxepine [60], eszopiclone [61] and sodium oxybate [61] were considered to have an acceptable risk. Lately, some Chinese medicines also show promise in RCTs, we ought to use them discreetly based on the severe side effects, such as heart failure [53].

Melatonin is another pharmacologic therapy option for insomnia and has been studied in patients with PD. In a randomized, double-blind trial, 40 PD patients were treated with melatonin, compared with a matched placebo. The authors found significant improvement in subjective sleep quality [54]. There were no adverse events and the melatonin therapy had no impact on motor function [54]. The lack of objectively measured improvement was broadly consistent with the findings from meta-analyses of melatonin for sleep in non-PD populations, which suggested a statistically significant, but clinically modest, improvement in objectively measured sleep [62]. Individual patients, however, may appreciate a symptomatic improvement in sleep quality, even in the absence of objective improvement. Lately, two studies showed that melatonin had a statistically significant improvement in both subjective and objective sleep quality [45,55]. Two–10 mg of Melatonin significantly improved overall sleep quality, besides there were no safety concerns observed in the trials on melatonin for the treatment of insomnia in PD [63]. Thus, melatonin could be considered as a more effective treatment for insomnia in patients with PD.

**Table 2 antioxidants-12-00396-t002:** Insomnia in Parkinson’s disease and their management supported by randomized controlled trials.

Method		Subjects (Dose/Duration)	Findings/Status	Study (Reference)
**Non-melatonin based management**				
Dopamine agonist	Ropinirole	347 PD patients (2–24 mg, 24 weeks)	Showed no statistical difference to the placebo in the PD sleep scale.	Clinical trials (Li SH, 2013 [53])
	Rotigotine	287 PD patients (16 mg, 12 weeks)	Significant effects on sleep quality and maintenance, with adverse events, such as nausea, application site reactions and dizziness.	Clinical trials (Trenkwalder C et al., 2011 [59])
	Apomorphine	46 PD patients (5 mg/h, 54 days)	Improved scores of insomnia and self-estimated clinical global impression of sleep quality, as associated with more frequent dizziness.	Clinical trials (De Cock VC, 2022 [64])
	Pergolide	30 PD patients (1 mg, 6 weeks)	Nighttime pergolide worsened sleep activity and was associated with an increased frequency of adverse events.	Clinical trials (Comella CL, 2005 [65])
CBT		77 PD patients (/, 10 weeks)	Symptoms of insomnia were significantly lower after treatment. Whereas, participants with difficulties using the technology associated with treatment were excluded.	Clinical trials (Kraepelien M, 2020 [65])
		18 PD patients (/, 6 weeks)	Reduced the ISI and the CGI-C, however, it was costly and required trained clinicians to administer.	Clinical trials (Rios Romenets S, 2013 [60])
Chinese medicine	YXQN	61 PD patients (4 g, 12 weeks)	Improved the PD sleep scale, but with some complaints of a bitter taste	Clinical trials (Pan W, 2013 [66])
	GPG	121 PD patients (6 g, 6 months)	Symptoms of insomnia were significantly lower after treatment with some severe side effects, such as pruritus and heart failure.	Clinical trials (Zhao GH, 2013 [6])
Eszopiclone		30 PD patients (2/3 mg, 6 weeks)	Significant improvement in the number of awake periods and quality of sleep, whereas, 33% of patients reported adverse events, such as dizziness.	Clinical trials (Menza M et al., 2010 [67])
Doxepin		18 PD patients (10 mg, 6 weeks)	Improved the ISI and the PSQI disturbances subscale, 3 patients taking doxepin reported mild fatigue, transient mild nausea and orthostatic dizziness, respectively.	Clinical trials (Rios Romenets S, 2013 [60])
Sodium oxybate		12 PD patients (3–9 g, 6 weeks)	Increased SWS and improved subjective nighttime, but it induced OSA in 2 patients and parasomnia in 1 patient.	Clinical trials (Büchele F, 2018 [61])
**Melatonin based management**				
Melatonin		51 PD patients (10 mg, 12 weeks)	The melatonin-treated group had a significant improvement of subjective sleep quality, compared with the placebo group.	Clinical trials (Daneshvar Kakhaki et al., 2020 [55])
		40 PD patients (2 mg, 4 weeks)	The melatonin-treated group had an improvement of in the total sleep time.	Clinical trials (Medeiros CA et al., 2007 [54])
		40 PD patients (5/50 mg, 10 weeks)	Significant improvement in the subjective sleep disturbance and sleep quantity was observed during a 5 mg melatonin treatment, compared with the placebos.	Clinical trials (Dowling GA et al., 2005 [68])
		38 PD patients (3 mg, 6 weeks)	The melatonin-treated group had significant changes in the total point scores on the PSG at the end.	Clinical trials (Litvinenko et al., 2012 [46])
		30 PD patients (2 mg, 4 weeks)	In the melatonin-treated group, PSQI improved, compared to the control group.	Clinical trials (Ahn et al., 2020 [45])
**Differences**				
Melatonin is more effective with fewer side effect.				

PD, Parkinson’s disease; RCT, randomized controlled trial; PSQI, Pittsburgh sleep quality index; CBT, cognitive behavioral therapy; ISI, insomnia severity index; CGI-C, clinical global impression of change; SWS, slow-wave sleep time; OSA, obstructive sleep apnea; YXQN, Yang-Xue-Qing-Nao; GPG, guling pa’an granule.

### 3.3. Sleep Related Breathing Disorders

SBDs are characterized by abnormalities in respiration during sleep [69]. Disorganized respiration with frequent central and obstructive apneas was found in patients with parkinsonism [70]. In prospectively evaluated studies, 15–76% of PD patients had moderate-to-severe obstructive sleep apnea (OSA), despite a normal body mass index [71]. OSA can lead to various cardiovascular and neurologic consequences with a negative impact on the quality of life in patients with PD [72]. Specifically, repeated hypoxemic events from OSA may contribute to CNS inflammation and oxidative injury, both of which can contribute to cognitive dysfunction [72]. An abnormal tone of the muscles surrounding the upper airway has been suggested as contributing to sleep-disordered breathing in PD [73]. Treatments for PD-SBD patients are the same as for the general population together with continuous positive airway pressure (CPAP), surgery, oral appliances, postural therapy and weight loss [72]. In past years, two RCTs have demonstrated that using CPAP to treat OSA in PD patients significantly improves objective measures, such as nocturnal oxygenation [74,75] (Table 3). Whereas the associated side effects are apparent, including dental injury in the case of oral appliance, poor tolerance and adherence [76].

Although there are no randomized trials investigating the treatment of melatonin specifically in the PD-SBD population, melatonin has been shown to ameliorate the complications caused by SBDs in both animal models and clinical trials. Kaminski RS et al. found that melatonin prevents the well-recognized increase in glucose levels that usually follows exposure to intermittent hypoxia in animal models of sleep apnea [77]. The protection conferred by melatonin may be related to its antioxidant mechanism. By human trials, it is proved that OSA aggravates EDS as the Epworth sleepiness scale (ESS) score of PD patients with OSA was higher than that of those who did not [78]. OSA may accelerate neuroinflammation through intermittent hypoxia, causing neuronal damage in brain regions that promote arousal, and eventually lead to PD-EDS, a condition that melatonin may treat.

### 3.4. Excessive Daytime Sleepiness

EDS is an another common problem of sleep dysfunction among PD patients who show the unquenchable need for sleep or inadvertent descent into drowsiness or sleep during daytime due to the inability to stay awake [79]. Due to the differences between the supervisory and objective methods to measure sleepiness and demographic characteristics, the prevalence of EDS symptoms in PD patients was 20–60% per statistical analysis of the previous literature, which had such a wide span [80]. Similar to RBD, EDS is also considered to be one of the prodromal features of PD and controversially, it has also been described as a symptom present in the advanced stage of PD. In addition, RBD patients with EDS have a higher possibility to develop PD, especially in older patients [81].

For the treatment of EDS, several therapies have been evaluated among PD patients (Table 4). According to their results, the first-line therapies for hypersomnia include bright light therapy, caffeine and the stimulant modafinil [72]. Bright light therapy may be effective in preventing EDS but there is no consensus on the optimal phototherapy parameters. Further research is needed to determine its optimal timing, dosage and treatment duration. The extents of effect of caffeine on EDS in PD patients are still unclear, and its clinical effect gradually lessens over time [82]. Modafinil is well tolerated in EDS patients, but there are side effects, such as headache, nausea, dry mouth and anorexia [83]. The effect of modafinil on the cardiovascular system, including the increase in blood pressure and heart rate [84], should be noted for some elderly PD patients with serious cardiovascular disease.

Melatonin has shown a role in treating EDS symptoms in PD patients. Similarly, Dr. Dowling employed a multi-site, double-blind, placebo-controlled and crossover trial involving 40 patients with PD [68]. This study showed that compared with taking a placebo, the patients taking 5 mg of melatonin had significantly improved daytime sleepiness in the weekly measurement of daytime sleepiness score according to the general sleep disorder scale (GSDS), but this is inconsistent with the results obtained through the ESS score [85]. The authors analyzed that the cause may be that the GSDS evaluated the patient’s sleepiness across a week while ESS for a day, thus the outcomes of the sleepiness measure are slightly different. Recently, a study reflected that melatonin significantly improved the ESS scores of PD patients, compared with a placebo [43], further certifying melatonin’s therapeutic effects on excessive sleepiness. More studies are needed to evaluate the effect of melatonin on EDS with PD by using different scales. Compared with PD patients without EDS, the magnitude and total amount of melatonin secretion in patients with EDS are significantly reduced [86]. Melatonin is an endogenous factor that affects the circadian rhythm, indicating that EDS is closely related to the disorder of circadian rhythm. It has been confirmed that melatonin can improve the symptoms of EDS by improving the circadian rhythm in other neurological diseases [87], but it is unclear how melatonin secretion is reduced in PD patients.

**Table 4 antioxidants-12-00396-t004:** EDS in Parkinson’s disease and their management supported by randomized controlled trials.

Method	Subjects (Dose/Duration)	Findings/Status	Study (Reference)
**Non-melatonin based management**			
Modafinil	40 PD patients (200–400 mg, 4 weeks)	Failed to significantly improve EDS in PD, compared with a placebo.	Clinical trials (Ondo WG et al., 2005 [88])
	21 PD patients (200 mg, 3 weeks)	Modestly effective for the treatment of daytime sleepiness with side effects, such as headache, nausea, dry mouth and anorexia.	Clinical trials (Adler CH et al., 2003 [89])
	12 PD patients (100/200 mg, 2 weeks)	Significant improvement in the ESS score, compared to a placebo; side effects with modafinil included insomnia, constipation, diarrhea and dizziness.	Clinical trials (Högl B et al., 2002 [90])
Caffeine	121 PD patients (200 mg, 18 months)	Slight improvement in sleepiness over the first 6 months, which attenuated over time.	Clinical trials (Postuma RB et al., 2017 [82])
Trazodone	112 PD patients (50 mg, 4 weeks)	Significant improvement in ESS and side effects were reported in two patients, one dizziness and one orthostatic hypotension.	Clinical trials (Hadi F, 2022 [43])
Piribedil	80 PD patients (100–30 mg, 11 weeks)	Reduced daytime sleepiness with low ESS scores, nevertheless, worsening of PD symptoms, falls, insomnia, dizziness, edema and nausea were observed.	Clinical trials (Eggert K, 2014 [91])
Solriamfetol	66 PD patients (75/150/300 mg, 4 weeks)	Safety and tolerability were consistent with its known profile. There were no significant improvements on ESS.	Clinical trials (Videnovic A, 2021 [92])
Bright light therapy	31 PD patients (Bright/dim-red light, 2 weeks)	Significant improvements in sleepiness assessed by the ESS, however, there is no consensus on the optimal timing, dosage and duration for it.	Clinical trials (Videnovic A et al., 2017 [93])
tDCS	23 PD patients (2.1 mA, 2 weeks)	Significant effect on fatigue but no effect on daytime sleepiness reduction.	Clinical trials (Forogh B, 2017 [94])
Sodium oxybate	12 PD patients (3–9 g, 6 weeks)	Substantially improved EDS as measured objectively and subjectively, but it induced OSA in two patients and parasomnia in one patient.	Clinical trials (Büchele F, 2018 [61])
**Melatonin based management**			
Melatonin	112 PD patients (3 mg, 4 weeks)	Significant improvement in ESS and no adverse effects were reported in the melatonin group.	Clinical trials (Hadi F et al., 2022 [43])
	40 PD patients (5/50 mg, 10 weeks)	Melatonin had significantly improved daytime sleepiness in the measurement of daytime sleepiness score in GSDS.	Clinical trials (Dowling GA et al., 2005 [68])
	40 PD patients (2 mg, 4 weeks)	No statistically significant difference between melatonin and placebo groups regarding the ESS decreasing.	Clinical trials (Medeiros CA et al., 2007 [54])
	38 PD patients (3 mg, 6 weeks)	ESS scores were slightly increased after treatment with melatonin.	Clinical trials (Litvinenko et al., 2012 [46])
	30 PD patients (2 mg, 4 weeks)	No statistically significant difference between melatonin and placebo groups regarding the ESS decreasing.	Clinical trials (Ahn et al., 2020 [45])
	28 PD patients (25 mg, 3 months)	There were fewer patients who reported sleeping poorly after the MEL intervention although the presence of abnormal daytime and nighttime sleepiness continued.	Clinical trials (D L Delgado-Lara et al., 2020 [95])
**Differences**			
Melatonin may improve somnolence in PD patients with milder adverse effects.			

EDS, excessive daytime sleepiness; PD, Parkinson’s disease; RCT, randomized controlled trial; ESS, Epworth sleepiness scale; GSDS, general sleep disturbance scale; tDCS, transcranial direct current stimulation.

### 3.5. Restless Legs Syndrome and Periodic Limb Movements

RLS is a sensory-motor disorder characterized by discomfort of and urge to move the legs, primarily during rest or inactivity, partial or total relief with movement, with presence or worsening exclusively in the evening [96]. Reported prevalence of RLS in the PD population varies widely between 8% and 80% [1,97]. It is often associated with PLMS, sudden jerking movements of the limbs which occur involuntarily during sleep and of which the affected individuals may remain unaware [98]. Other studies suggest that PD patients with RLS are older at PD onset, more advanced PD stages, severe limb parkinsonism, depression, anxiety, dysautonomia and worse nutritional status [99]. RLS may be the underlying cause of insomnia, such as difficulty in sleep initiation too. 

Randomized clinical trials examining the management of RLS, specifically in the PD population are rare, only one trial with small sample sizes has been conducted with deep brain stimulation (DBS) [94]. In a single-center, double-blind, randomized and crossover trial, DBS of the subthalamic nucleus (STN) and SNs reticulata improved nocturnal RLS symptoms in PD patients and was more effective than conventional subthalamic stimulation [100] (Table 5). However, stimulation induced side effects, such as uncomfortable feeling and increased confusion and hallucinations and aggressiveness [101]. Additionally, RLS may recur after STN-DBS [101]. Although several dopaminergic drugs have also been independently proven to be effective in treating RLS symptoms in non-PD patients in randomized trials, it is necessary to confirm whether these medications have an acceptable risk [102].

Dr. Whittom conducted an eight RLS subject study in three conditions: at baseline, after administration of melatonin and during bright light exposure [103]. The severity of RLS symptoms was assessed by the suggested immobilization test (SIT). Analyses showed a significant increase in the SIT-PLM index when subjects received exogenous melatonin, compared to both baseline and bright light conditions. It implied that motor symptoms worsened during the SIT when subjects received exogenous melatonin [103]. Considering the small sample size, more RCTs are essential to determine the efficacy of melatonin, also paying attention to melatonin’s effects on motor symptoms in PD.

## 4. Possible Neuroprotective Mechanisms of Melatonin Therapy for Sleep Disorders in PD

Melatonin can easily cross the blood–brain barrier and plays a major role in a variety of neuroprotective functions, such as antioxidant, anti-inflammatory, anti-excitotoxity, anti-synaptic dysfunctions, anti-apoptotic activities and restoration of the circadian rhythm. Likewise, we focus on sleep disorders in PD and try to explore the possible neuroprotective mechanisms of melatonin therapy in six categories (Figure 1).

### 4.1. Restoration of the Circadian Rhythm 

Circadian rhythm sleep–wake disorders are characterized by a chronic or recurring sleep disruption owing to an alteration of the circadian system or a misalignment between the endogenous circadian rhythm and lifestyle sleep–wake schedules [79]. Multiple impaired brain areas or neurotransmissions in PD may affect the input to the hypothalamic supraschiasmatic nucleus (SCN), the master clock of the endogenous circadian rhythm [104]. This effect can lead to significant changes in sleep and wake times among PD patients. Mechanistically, SCN with PD is regulated by the altered expression of core genes, which are known as the “clock genes”. Such genes encode a variety of transcription factors that rhythmically regulate the expression of downstream genes and affect circadian rhythms [105]. The key clock genes include the circadian locomotor output cycle kaput (*Clock*), brain and muscle aryl-hydrocarbon receptor nuclear translocator-like 1 (*Bmal1*), Period 1 (*Per1*), Period 2 (*Per2*), Period 3 (*Per3*) and cryptochrome (*Cry 1* and *Cry 2*) [104]. Generally, the levels of *Per* and *Cry* genes expression were lowest early in the morning and peaked approaching midnight [106]. The Clock/Baml1 proteins initiate the transcription of *Per* and *Cry*, then these target genes accumulate as a result [104]. In return, Per and Cry proteins dimerize and inhibit the transcriptional activity of Clock/Baml1 proteins, resulting in the downregulation of their own expression [104]. Such transcription-translation feedback loop forms a 24-h oscillation form, maintaining the normal circadian rhythm of the organism [107] (Figure 2). The abnormal expression level and dysfunction of the clock genes can lead to the disturbance of the circadian rhythm, which has a significant impact on the development and progression of PD [105]. For example, in a gene polymorphism analysis, Lou et al. documented that the scores of PQSI in *Clock* variant carriers in PD patients were significantly higher, compared with non-carriers [108]. The results suggested that the *Clock* variation was an independent risk factor for sleep disorder symptoms in PD. In a gene expression analysis, PD patients were found to have increased sleep latency, decreased sleep efficiency, decreased REM sleep and lack of time-dependent changes in the *Bmal1* expression, especially in the early stage of the disease [109]. Furthermore, an altered *Bmal1* expression was associated with the diminution of circulating melatonin. 

The use of melatonin, a chronotherapy in PD, is the majority of ongoing clinical trials. Notably, melatonin receptors are present in the SCN so that circulating melatonin can deliver feedback to the SCN. The circadian clock is affected by exogenous melatonin, implying that melatonin supplement is a promising therapy method [110]. There were two studies investigating the effect of melatonin on the changes in the clock genes’ expressions, including an animal experiment and a randomized, double-blind, crossover clinical trial. The animal experiment used the rotenone-induced PD (RIPD) male Wistar rat model and revealed that the daily rhythm of *Per1* was delayed by about 6 h in the RIPD rat model [107]. Subsequent administration of melatonin restored the *Per1* daily rhythm. Differently, results suggested that melatonin increased the Baml1 levels but not Per1 levels in PD patients [95]. Furthermore, this showed that there were reduced changes in body gesture during nighttime, improved sleep efficiency and decreased the daytime sleepiness with melatonin intake. 

### 4.2. Antioxidant Mechanism 

Smith AM et al. isolated the peripheral blood mononuclear cells from PD patients, PD-RBD patients vs. age-, sex-matched controls for clinical evaluation [111]. They found the level of mitochondrial dysfunction and oxidative stress in the peripheral blood mononuclear cells of PD patients increased, especially the level of mitochondrial reactive oxygen species and decreased the superoxide dismutase regardless of RBD. Moreover, a colon biopsy showed that PD subjects with RBD had significantly increased oxidative damage measured by lipid peroxidation in intestinal tissues, compared with the control group. Hence, the evidence suggests that RBD is closely related to oxidative stress-related damage in PD. Sleep fragmentation insomnia is one of the most common types of insomnia in PD patients. In mice models, sleep fragmentation leads to learning deficits. It was found that this was related to the increased gene expression and activity of NADPH oxidase in hippocampus and the cortex of wild type mice [112]. Reversely, mutant mice lacking NADPH oxidase activity were refrained from learning deficits, indicating that sleep fragmentation appears to induce oxidative stress. It is concerned that OSA can lead to neural injury, particularly OSA with intermittent hypoxemia related to oxidative damage [113]. There is evidence that the treatment of OSA can partially reverse the dysfunction [114] which is instructive for the treatment of PD patients with OSA. NADP oxidase and inducible nitric oxide synthase (iNOS) were found to mediate this injury and the associated proinflammatory response [69].

Without any doubt, melatonin is one of the most powerful antioxidants acting at various levels, from direct radical scavenging and enzymatic regulation of oxidant formation to mitochondrial radical avoidance [115]. Melatonin prevents neuronal apoptosis triggered by ROS and maintains glutathione homeostasis. Moreover, it also prevents a loss in mitochondrial complex I activity, which may be a potential area in treating neurological disorders associated with oxidative stress [58]. The antioxidative protection of melatonin is not limited to radical scavenging, it exerts several actions which collectively contribute to the prevention of oxidative damage [18]. The antagonism of melatonin, using methods, such as constant light and pinealectomy, alleviates the symptoms during advanced stages of the disease in animals and humans [116]. 

In addition, free radicals may be generated during enzymatic reaction and non-enzymatic reaction in the body and the generation and elimination of free radicals in the normal body are in dynamic balance. Once this balance is broken, free radicals would cause damage to biological macromolecules, such as lipids, proteins and nucleic acids, resulting in the destruction of cell structure. Melatonin protects the cell structure, prevents DNA damage and reduces the amount of peroxide in the body by scavenging free radicals, anti-oxidation and inhibiting lipid peroxidation. A remarkable result of this study is that N1-acetyl-5-methoxykynuramine (AMK), a brain metabolite of melatonin, which was as efficient as melatonin treatment in counteracting iNOS production, oxidative stress and mitochondrial dysfunction induced by MPTP [12].The antioxidant effects of melatonin and the immune-pineal axis are both well established. Recent studies have shown that the immune-pineal axis acts as an immunological buffer, neurohormonal switch and it also intricately links the pathogenesis of neurodegenerative diseases and inflammation at a molecular level [117].

Moreover, melatonin can correct the expression of cytokines including fibroblast growth factor 9 (FGF9) and glial cell-derived neurotrophic factor (GDNF). FGF9 is expressed by neurons and glial cells to support the survival of neurons, as well as to increase glial cell proliferation. Melatonin prevents MPP+-induced FGF9 down-regulation to achieve its neuroprotective effects in vivo and in vitro, although melatonin-only treatment did not up-regulate the FGF9 expression [118], suggesting melatonin attenuates MPP+’s action. GDNF is a potent survival factor for dopaminergic neurons in the CNS [119]. Melatonin attenuates the pathological phenomena that GDNF expression compensatorily increased in the contralateral striatal, supporting a physiological role for melatonin in correcting the expression of growth factors, which is normally defective in PD [120].

### 4.3. Anti-Inflammatory Mechanism

Inflammation is speculated to have a role in the initiation and progression of neurodegenerative diseases [121]. Systemic inflammatory factors increase in the body with the process of aging which leads to chronic activation of microglia and parenchymal macrophages in the CNS with an increased number of astrocytes [104]. In animal experiment studies, the levels of the inflammatory factor IL-1β were increased in the hypothalamus of REM sleep deprived mice, suggesting a possibility for neuroinflammation in sleep disorders [122]. 

Hu Y et al. found that the levels of α-synuclein, NO, IL-1β and TNF-a were increased, compared with PD patients without RBD symptoms in CSF [123]. The elevated α-synuclein oligomer levels result in increased microglial activation. Degenerative or dead neurons may release more α-synuclein oligomers into the extracellular space of RBD related areas which may lead to persistent microglial activation and neuronal death in PD patients. Correspondingly, it will probably aggravate the inflammation in the central and peripheral systems and cause the occurrence of RBD eventually. Chronic sleep fragmentation has also been found to selectively increase the cortical expression of TNF-α [124]. Correspondingly, the TNF-α receptor complete knockout mice and the mice treated with neutralizing antibody showed learning defects related to sleep fragmentation, indicating that insomnia appears to induce inflammation. In addition, intermittent hypoxia seems to induce oxidative stress and inflammation, just as sleep fragmentation does. For OSA patients, intermittent hypoxemia also induces proinflammatory transcription factor NF-κB and causes systemic inflammatory reaction [125]. In addition, The elevated plasma levels of C reactive protein, TNF-α, IL-6 and IL-8 in OSA indicate the existence of systemic inflammation [126]. In this condition, these OSA-related mechanisms might theoretically lead to neuroinflammation, promoting neurodegeneration and aggravating the progress of PD. Studies show the objective sleepiness assessed by the MSLT scale is related to the higher apnea hypopnea index of PD patients [80]. The recurrent process of airway collapse and obstruction in patients with OSA leads to apnea and subsequently promotes the development of inflammation [121]. Likewise, it induces the damage of neurons in several wake-promoting brain regions, such as the gray matter around the dopaminergic ventral aqueduct and noradrenergic blue spot, leading to EDS in PD patients [127]. Therefore, impaired wake-promoting brain regions may be the common pathogenesis of EDS and other sleep disorders in PD patients [80], and neuroinflammation caused by OSA may aggravate this pathological process.

The anti-inflammatory and neuroprotective effects of melatonin play an important role in improving the prognosis of neurodegenerative diseases, such as PD [117]. As reported, melatonin blocks iNOS synthesis in the macrophages and microglia by inhibiting the NFκB pathway, thereby limiting its influence on local inflammatory responses to some degree. Other related signaling pathways include the up-regulation of nuclear factor erythroid 2-related factor 2 and of the Toll-like receptor-4 activation and high-mobility group box-1 signaling receptors, and prevention of the NLRP3 inflammasome activation by melatonin [128]. Moreover, melatonin inhibits the expression of cytokines and adhesion molecules, as well as reactive oxygen and nitrogen species [129]. In addition, melatonin has a profound regulatory effect on microglia/macrophage polarization toward the anti-inflammatory phenotype. The latest study proved that melatonin regulated microglia’s shift towards an anti-inflammation phenotype to alleviate neuroinflammation and retinoic acid receptor-related receptor alpha (RORα) exerts a key role in the regulation of microglial polarization [130].

### 4.4. Anti-Excitotoxicity Mechanism

RBD and EDS are the common sleep disorders among the prodromal symptoms of PD [8]. These two disorders may share the mechanism basis caused by the activity changes of two state regulated brain stem nuclei located in the pontine tegmentum: the laterodorsal tegmental nucleus (LDT) and the pedunculopontine tegmental nucleus (PPT) [131]. Cholinergic neurons within the LDT and PPT can regulate arousal and maintain wakefulness, and modulate REM sleep initiation [132]. In LDT and PPT, the monomeric form of α-synuclein induces excitability and increases calcium influx, which increases its excitotoxicity and damages sleep-control nuclei [133].

Melatonin displays an anti-excitotoxic activity, which was associated with the inhibition of calcium influx and NO release [115]. Meanwhile, it possesses strong anticonvulsant properties to counteract efficiently the actions of various excitotoxins [134]. Moreover, melatonin inhibits the α-synuclein assembly and attenuates kainic acid-induced neurotoxicity and arsenite-induced apoptosis. Melatonin also impaired the augmented expression of α-synuclein protein in dopamine-containing neurons. It blocks the α-synuclein fibril formation and destabilizes preformed fibrils by inhibiting protofibril formation and secondary structure transitions, reducing the α-synuclein cytotoxicity. In addition to the pronounced anti-excitatory and anti-excitotoxic effects, melatonin prevents neuronal death induced by kainate, an ionotropic glutamate receptor agonist. As well, melatonin has been shown to reduce neuronal damage due to the toxicity of cadmium, hyperbaric hyperoxia, toxicity by δ-aminolevulinic acid, γ radiation, focal ischemia, brain trauma, and that resultant from several neurotoxins [135]. It is unknown whether these effects are attributable to MT1 and MT2 activities.

### 4.5. Anti-Synaptic Dysfuction

In the past, PD was considered to be a purely sporadic environmental disease, but it now it is known that there is a genetic link. Mutations of the leucine-rich repeat kinase2 (LRRK2) gene is one of the most common monogenic forms of PD [136]. It encodes a multi-domain dual-enzyme leading to cell death and neurodegeneration distinct from any other protein linked to neurodegeneration [137]. Ran et al. found that melatonin decreased LRRK2-induced memory dysfunction in the drosophila model of PD [138]. In addition, a study indicated that the mushroom bodies expression of LRRK2 resulted in sleep fragmentation and presynaptic dysfunction associated with the reduced frequency of EPSPs and increased the synaptic bouton density [139]. As well, melatonin significantly attenuated LRRK2-induced sleep problems and rescued the reduced cholinergic synaptic excitatory postsynaptic potential (EPSP) frequencies, without affecting the increase in synaptic bouton density. These results suggest that the beneficial effects of melatonin may be associated with the promotion of synaptic transmission.

### 4.6. Anti-Apoptotic Molecule

The pathogenesis of RLS in PD is related to the degeneration of dopaminergic neurons, hypofunction of iron metabolism and opioid system [140]. Withania somnifera is a plant-derived neuroprotective and dopaminergic agent. A case report showed that use of Withania somnifera completely alleviated the refractory RLS in an elderly female PD patient, which might be due to drug’s anti-apoptotic and dopaminergic effects [141]. This drug protects the dopaminergic neurons from apoptosis and the reduction of activated astrocytes by increasing B-cell lymphoma-2 (BCL-2) and free radicals scavenging, reducing BCL-2 associated X protein (BAX) [142]. Another study found that 100 μm exposure to desferoxamine for 48 h could cause the DNA fragmentation of DA neurons in SN of rats [143]. Administration of δ-opioid peptide [D-Ala2, D-Leu5] enkephalin (DADLE) could significantly protect SN cells from iron deficiency impairment. The apoptosis family gene P53 induced by DNA-damage, and the pre exposure of DADLE prevents this activation. This suggests that RLS in PD is related to the apoptosis of dopaminergic neurons.

Increased oxidative stress, decreased antioxidant enzymes, and mitochondrial dysfunction lead to apoptosis and cell death of neurons [144]. Melatonin (10 mg/kg) decreased DNA fragmentation in striatum and midbrain and lowered apoptosis cells in midbrain in PD rats induced by MPTP [145]. Neurodegeneration occurs in PD, at least in part, through the activation of the mitochondria-dependent apoptotic molecular pathway. In addition, dopamine is neurotoxic under certain pathological conditions and induces apoptosis via redox sensitive JNK activation [146]. It is reported that the action of melatonin to inhibit JNK signaling cascade diminishes the induction of the phosphorylation of c-Jun in MPP (+)-treated and 6-hydroxydopamine-induced SK-N-SH-cultured cells [147].

## 5. Conclusions and Future Perspectives

For melatonin, the ability to cross the blood–brain barrier and its short life with no significant side effects make it a promising neuroprotector to ameliorate sleep disorders for PD patients. This review discusses the application and possible mechanisms of melatonin in the sleep disorders of PD patients. However, these molecular mechanisms are only indicated to occur in the cerebral neurons, but it is not clear whether it occurs in the midbrain SN dopaminergic neurons concretely. Furthermore, although it demonstrated that melatonin affected the levels of clock genes, besides Baml1 and Per1, it lacks the measurement of other clock genes in PD patients and the altered expression levels after melatonin administration. In addition, despite that melatonin is relatively commonly used in the PD population, clinical trials and RCTs of melatonin in PD are scarce. Long-term data on the efficacy of melatonin in treating sleep disorders in PD are not available, though short-term studies are promising [148]. In fact, PD patients may carry more than one type of sleep disorder and the outcomes of melatonin in treating these comorbidities are also unclear. Our results identified that melatonin is effective for ameliorating RBD, insomnia, EDS, SBDs, RLS and PLMS without physical dependence and other significant adverse reactions. Mechanistically, melatonin may exert its therapeutical effects through restoring the circadian rhythm and playing the antioxidant, anti-inflammatory, anti-excitotoxity, anti-synaptic dysfunction and anti-apoptotic activities.

Presently, about eight clinical trials registered at clinicaltrials.gov and EU Clinical Trials Register are exploring the therapeutic effects of melatonin and its analogues on PD-concordant sleep disorders. Focusing on sleep disorders, two clinical studies had results. The first one (NCT02789592), to evaluate the effect of melatonin intake on the RBD symptoms in patients with PD, Ahn JH et al. performed a randomized placebo-controlled, double-blind, parallel-group clinical trial. They found that prolonged-released melatonin was an effective treatment for PD patients with RBD [45]. The second (NCT00745030) aimed to investigate safety and efficacy of ramelteon for the treatment of RBD in subjects with parkinsonism (Table 1). Unfortunately, the trial had results but terminated eventually because of the low subject recruitment and enrollment. Additionally, the other unfinished studies are currently ongoing but do not have any information available at present. In the future, we expect the outcomes of these trials and look forward to more clinical trials defining the therapeutic effects of melatonin on sleep disorders in PD.

## Figures and Tables

**Figure 1 antioxidants-12-00396-f001:**
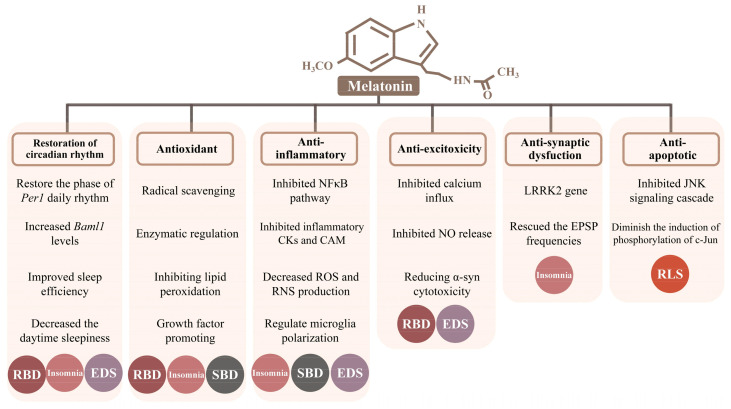
Six possible neuroprotective mechanisms of melatonin therapy for Parkinson’s disease. Noted: N-acetyl-5-methoxytryptamine, melatonin. Opening rectangles under the mechanism represent its involved specific molecular mechanisms, respectively. Different colored circles distinguish five types of sleep disorders. Circles, sleep disorders through the specific mechanisms. Abbreviations: Per1, Period 1; Baml1, brain and muscle aryl-hydrocarbon receptor nuclear translocator-like 1; CKs, cytokines; CAM, cell adhesion molecule; ROS, reactive oxygen species; RNS, reactive nitrogen species; α-syn, α-synuclein; LRRK2, leucine-rich repeat kinase2; EPSP, excitatory postsynaptic potential.

**Figure 2 antioxidants-12-00396-f002:**
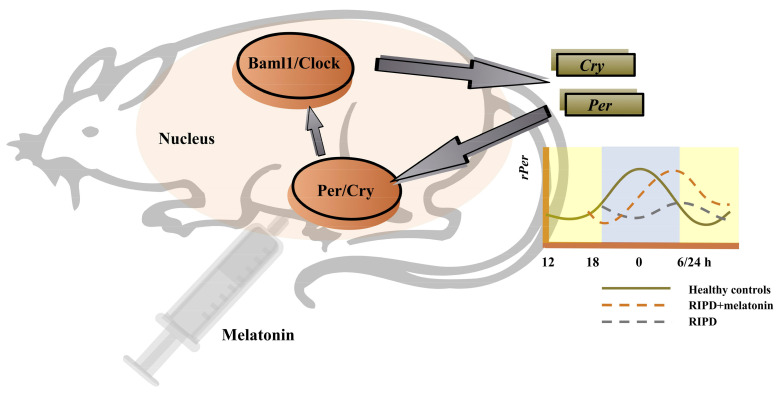
The regulation of the key clock genes and possibly the effect of melatonin on these genes. Noted: The levels of *Per* and *Cry* genes expression were lowest early in the morning and peaked approaching midnight. The Clock/Baml1 proteins initiate the transcription of *Per* and *Cry*, then these target genes accumulate as a result. In return, Per and Cry proteins dimerize and inhibit the transcriptional activity of the Clock/Baml1 proteins, resulting in the downregulation of their own expression. Such transcription-translation feedback loop forms a 24-h oscillation form, maintaining the normal circadian rhythm of the organism. Administration of melatonin could restore the *rPer1* daily rhythm which was delayed by about 6 h in the RIPD rat model. Abbreviations: RIPD, rotenone-induced PD; Bmal1, Brain and muscle aryl-hydrocarbon receptor nuclear translocator-like 1; Clock, Circadian locomotor output cycle kaput; Per, Period; Cry, Cryptochrome.

**Table 3 antioxidants-12-00396-t003:** SBDs in Parkinson’s disease and their management supported by randomized controlled trials.

Method	Subjects (Dose/duration)	Findings/Status	Study (Reference)
**Non-melatonin based management**			
CPAP	121 PD patients (/, 6 weeks)	Effective in reducing apnea events, improving oxygen saturation and deepening sleep in patients, but with poor adherence.	Clinical trials (Neikrug AB et al., 2014 [74])
	38 PD patients (/, 6 weeks)	Significant reduction of AHI levels, with great effort to help treatment adherence.	Clinical trials (Harmell AL et al., 2016 [75])
**Melatonin based management**			
Melatonin	72 Balb/c mice (0.2 mg/kg, 2 weeks)	Melatonin prevented the increase in glucose levels that usually followed exposure to intermittent hypoxia in animal models of sleep apnea.	Animal experiment (Kaminski RS et al., 2015 [77])
**Differences**			
Melatonin ameliorates sleep apnea more acceptably and is easier to adhere.			

SBDs, sleep-related breathing disorders; PD, Parkinson’s disease; RCT, randomized controlled trial; CPAP, continuous positive airway pressure; AHI, apnea–hypopnea index.

**Table 5 antioxidants-12-00396-t005:** Restless legs syndrome and periodic limb movement in Parkinson’s disease and their management supported by randomized controlled trials.

Method	Subjects (Dose/Duration)	Findings	Study (Reference)
**Non-melatonin based management**			
STN + SNr-DBS	15 PD patients (1.5–5.0 mA, 6 weeks)	It is superior to the control group in improving RLS symptoms at night. However, stimulation induced side effects, such as uncomfortable feeling and increased confusion and hallucinations, aggressiveness.	Clinical trials (Hidding U, 2019 [100])
**Melatonin based management**			
Melatonin	8 patients (3 mg, 3 h)	Motor symptoms worsened during the SIT when subjects received exogenous melatonin.	Clinical trials (Whittom S et al., 2010 [103])
**Differences**			
Inclusive due to limited trials			

PD, Parkinson’s disease; RCT, randomized controlled trial; STN + SNr-DBS, simultaneous deep brain stimulation of the subthalamic nucleus and substantia nigra pars reticulata; SIT, suggested immobilization test.

## Data Availability

Not applicable.

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
