# Peer review of "Neuroprotective Effect of Melatonin on Sleep Disorders Associated with Parkinson’s Disease"

_antioxidants, 2023, doi:10.3390/antiox12020396_

Round 1

Reviewer 1 Report

An article “Neuroprotective effects of melatonin on sleep disorders of Parkinson’s disease” by Hu X. et al. speaks of its possible significance. In fact it is only possible. Commonly, the review is intended to summarize the available results, compare different methods of treatment and evaluate their benefits in order to indicate possible ways to alleviate the suffering of patients with this serious ailment. Unfortunately current review often simply lists the available drugs without a sufficient analysis of their advantages and disadvantages in comparison with melatonin. The form in which data on the treatment of patients with Parkinson's is presented does not stand up to criticism. Table 1 completely confuses the reader since the logic of placing in parallel columns (apparently for comparison) specific cases of treatment is not clear and inconvenient for the reader. The situation is even worse with Table 2, which contains a lot of inconclusive and unfinished studies. The only way to improve the review is to reorganize Table 1 in a form of several tables summarizing the discussion of drugs and treatments in the particular section.  For example, section 3.1 REM sleep disorders, explains the therminology, describes traditional drugs and doses used and is terminated by the corresponding Table which compare common methods with treatment by melatonin. Thus the data presented in one large chaotic table will be spread among a number of tables associated with a precise Sections. These tables can also include reliable data from Table 2, that is, data from published or at least effective studies. In a separate phrase, it can be mentioned that in addition, a number of unfinished studies have been done.

As a significant drawback, carelessness in the design should also be noted. On page 1 line 27 in is duplicated. The same page lines 32, 35, 37, 40 the [ ] marking the reference are absent. Line 150 “give new a perspective’ should be changed to “give a new perspective” etc.  In the Discussion Section the expression “Mitochondrial radical avoidance” is unsuccessful and radical scavenging describes fully enough the action of melatonin. Also unsuccessful is “decrease expression of ROS and RNS” and “decreased ROS and RNS production” fits the situation much better. 

Reviewer 2 Report

The authors focused on the sleep disorders present in Parkinson's disease, summarizing the neuroprotective effects of Melatonin.

There are missing spaces between words and references. There are numbers next to some words for example "cell death3 line 35, degree4 line37, ones5 line 40, response71 line 434.

The introduction presents the problematic of the review well and gives enough information to make it understandable.

Line 73, the word "bottommost" is missing a space.

Line 99: this word "rCry1" is not understandable. 

Figure 1 is helpful in summarizing the discussion. The fact that the sleep disturbances associated with each property are indicated is a good thing.

It would be useful to have a diagram showing the time of expression of the key clock genes during the day, their regulation and the "relationships" between them, and possibly the effect of melatonin on these genes.

Reviewer 3 Report

In this manuscript the authors review the information about the neuroprotective effect of melatonin on sleep disorders of Parkinson’s disease.

This review is interesting and gives a good overview of the topic. However, this manuscript needs  improvements and corrections mainly concerning the presentation of the results before publishing may be possible.

General points:

English language correction by native speaker is required. 

 Please add a list of abbreviations before References section to your manuscript.

 Please check and correct the citation style of your manuscript.

 Please do your List of references exactly according to “Antioxidants”.

Special points:

 Please add a Conclusion and Future perspectives sections to your manuscript.

 Introduction

 Lines 27-56: please add multiple references at the end of each of these sentences.

 Lines 32-40: Check citations (numbers

Main body of the manuscript

 Lines 67-88: please add multiple references at the end of each of these sentences.

 Lines 115-128:  please add multiple references at the end of each of these sentences.

 Lines 139-144: please add multiple references at the end of each of these sentences.

 Lines 153-158: please add multiple references at the end of each of these sentences.

 Table 1: please present this Table in better quality and better reading design. Please show a non-melatonin-based management and melatonin-based management as a separately Tables. Show clearly which citation belongs to which of the five sleep disorders. Always use an identical font.

Lines 179-180: please add multiple references at the end of this sentence.

Lines 216-223: please add multiple references at the end of each of these sentences.

 Lines 235-247: please add multiple references at the end of each of these sentences.

 Lines 260-274: please add multiple references at the end of each of these sentences.

 Lines 327-332: please add multiple references at the end of each of these sentences.

 Lines 336-348: please add multiple references at the end of each of these sentences.

 Lines 357-361: please add multiple references at the end of each of these sentences.

 Figure 1: please add the appropriate references for each part of this Figure.

Lines 387-394: please add multiple references at the end of each of these sentences.

 Lines 470-473: please add multiple references at the end of each of these sentences.

 Lines 517-518: please add multiple references at the end of this sentence.

Round 2

Reviewer 1 Report

The authors revised and significantly improved the manuscript. All my suggestions were taken into account. Now the review may be published. 

Reviewer 2 Report

The authors responded positively to all my comments and requests. 

Reviewer 3 Report

The manuscript has really improved and all my suggestions were fulfilled.